# Point-of-Care Strategies for Detection of Waterborne Pathogens

**DOI:** 10.3390/s19204476

**Published:** 2019-10-16

**Authors:** Sandeep Kumar, Monika Nehra, Jyotsana Mehta, Neeraj Dilbaghi, Giovanna Marrazza, Ajeet Kaushik

**Affiliations:** 1Department of Bio and Nano Technology, Guru Jambheshwar University of Science and Technology, Hisar-Haryana 125001, India; ssmonikanehra@gmail.com (M.N.); jyotsanamehta1@gmail.com (J.M.); or; 2Department of Chemistry “Ugo Schiff”, University of Florence, Via della Lastruccia 3, 50019 Sesto Fiorentino, Florence, Italy; giovanna.marrazza@unifi.it; 3Department of Natural Sciences, Florida Polytechnic University, Lakeland, FL 33805-8531, USA

**Keywords:** POC sensor, smart sensing, health quality, water health diagnostics, nanotechnology

## Abstract

Waterborne diseases that originated due to pathogen microorganisms are emerging as a serious global health concern. Therefore, rapid, accurate, and specific detection of these microorganisms (i.e., bacteria, viruses, protozoa, and parasitic pathogens) in water resources has become a requirement of water quality assessment. Significant research has been conducted to develop rapid, efficient, scalable, and affordable sensing techniques to detect biological contaminants. State-of-the-art technology-assisted smart sensors have improved features (high sensitivity and very low detection limit) and can perform in a real-time manner. However, there is still a need to promote this area of research, keeping global aspects and demand in mind. Keeping this view, this article was designed carefully and critically to explore sensing technologies developed for the detection of biological contaminants. Advancements using paper-based assays, microfluidic platforms, and lateral flow devices are discussed in this report. The emerging recent trends, mainly point-of-care (POC) technologies, of water safety analysis are also discussed here, along with challenges and future prospective applications of these smart sensing technologies for water health diagnostics.

## 1. Introduction

Water is the most essential component for subsistence of life on the planet. The water resources worldwide are drowned due to a variety of pollutants, either biological or nonbiological. As per the World Health Organization (WHO) reports 2019 [1], more than one billion people are unable to access clean water sources. The water contaminants, i.e., physical, chemical, and biological substances, beyond certain levels may be harmful [2,3,4]. Globally, approximately 1.6 million people pass away yearly from waterborne diseases caused by biological contaminants. Such contaminated water is continuously affecting younger children (under the age of five years) severely. Most biological contaminants are organisms including bacteria, viruses, protozoans, and helminths. These microorganisms can enter human body through direct consumption of contaminated water and, as a result, cause adverse health risks.

Various countries have their legally enforceable standards for public water systems that are intended to limit contaminant levels in drinking water to ensure public health protection. As per WHO regulations regarding bacterial contaminants in drinking water, the total number of general bacteria and coliforms should not be detectable in any 100 mL volume of water. It has been observed through various reports that the risk of waterborne infection increases with the level of biological contamination. The major biological water contaminants are pathogens e.g., *Campylobacter, Clostridium, Salmonella, Staphylococcous,* colon bacillus, enteroviruses, human caliciviruses, *Entamoeba histolytica, Giardia lamblia,* microsporidia spp., *Anabaena, Microcystis, Schistosoma mansoni,* and *Taenia saginata.* These bacteria infect water rapidly. There are various pathways of pathogen entry inside the human body, either directly through drinking water or indirectly through food [5,6]. Further, these can be released into environment with excreta, knowingly or unknowingly, thereby contaminating lakes, rivers, and reservoirs.

Many incidents caused by water-borne diseases, such as cholera and dysentery, claimed several lives from the 19th to the 20th century [7]. In 1991, a major cholera outbreak accounted for large number of deaths in America and Africa. The reason behind this outbreak was poor wastewater treatment systems that led to *Vibrio cholera* contamination. In recent years, *Escherichia coli* (*E. coli*) O157 and other coliform bacteria have been found to be the major causes of water-borne infections [8]. *E. coli* O157 produces verotoxin, which may cause hemorrhagic colitis, thrombocytic thromhemolytic uraemia, and hemolytic uraemia syndrome [9,10]. Therefore, monitoring of biological contaminants is of utmost importance to clean the drinking water and eliminate the life-threatening diseases associated with polluted water. Conventional detection methods for biological contaminants include microbiological [11], nucleic acid amplification and hybridization [12], protein signature analysis [13], and immunological assays [14]. Some commercial devices, such as MicroChemLab, fatty acid methyl esters (FAME) analyzer, etc., are also available based on conventional detection methods. Although these conventional methods are accurate and highly sensitive, the instrumentation is bulky and the process requires trained personnel and sample pretreatment. Moreover, these devices cannot be applied for on-site detection of contaminants. Therefore, miniaturized, easy to operate, and affordable point-of-care (POC) devices exhibiting high specificity and sensitivity facilitating on-site detection are attracting research orientation.

The present, careful and critical, review recapitulates reported water contaminant detection platforms for water quality assessment. Conventional methods are summarized with scientific discussion associated with their limitations. Further, the advanced POC technologies for rapid and effective monitoring of biological contaminants are explored in terms of their different formats. Additionally, the challenges and future prospects of POCs for biological contaminants are also presented in this report to serve as a guideline to plan future research to develop next-generation sensors for water health assessment.

## 2. Conventional Methods to Detect Biological Contaminants

Water quality monitoring is the primary factor towards prevention of illness and infections caused by microorganisms. In the 20th century, due consideration has been given to water quality monitoring for pathogen detection. The various detection methods for biological contaminants are discussed here in detail (Figure 1). The conventional approaches are majorly focused on culture-based testing, separation and filtration of microbes, immunological methods, and nucleic acid-based detection. The continuous improvement in detection approaches can be observed, keeping in view the nature and origin of microorganisms to be detected.

### 2.1. Microbiological Assays Technique

The conventional microbiological methods rely on the growth and isolation of pathogens on the specific media. Some common microbiological diagnostics are based on microbiological cultures using medium (either selective or differential), Gram-staining, microscopy of specimens, and biochemical tests [15]. Culture-based microbiological assays, generally known as the golden standard, were the first developed methods for the qualitative and quantitative detection of biological contaminants i.e., pathogens [16]. The basic steps involved in the detection of pathogens via culturing are: enrichment, plating, incubation, colony counting, biochemical screening, and serological confirmation [17]. Culture-based methods are sensitive, inexpensive, and provide accurate information on the nature and number of microorganisms present in the sample [18]. However, these methods require several days to weeks to produce results, depending on the ability of the pathogen to multiply into visible colonies. Further, the steps of culture medium preparation, inoculation, and colony counting make the process labor intensive. Moreover, these methods have reached their limit in growing specific microorganisms in artificial media [19]. Despite these limitations, these conventional culture-based methods are still in clinical practice because they can provide diagnosis of acute infections in early stages in comparison to other technological advances.

### 2.2. Separation and Filtration Techniques

In order to minimize the time period required for culture-based plating assays, research efforts have been directed towards the development of alternatives for specific detection of pathogens and also to avoid chances of the false-negative results. In this regard, different separation and filtration techniques (i.e., chemical, physical, and antibody-based techniques) have been explored to recover and concentrate the microbes from contaminated water [20,21,22]. In order to spot the very low concentration of pathogens in water, an initial step of concentration or enrichment is required for their detection. After concentrating the target organism from a portion of a large quantity of material, a shortened and efficient detection of pathogenic microbes can be achieved. Further, membrane-based direct epifluorescent filtration technique (DEFT) is another approach for enumeration of microbes, depending upon the binding properties of the fluorescent dye. DEFT is a direct method to concentrate the target pathogens from a large sample volume [23]. These conventional approaches suffer with the use of harsh chemical reagents that can cause cell injury.

Another technique to shorten the isolation stage is immunomagnetic separation (IMS), in which the selective enrichment stage can be replaced with nongrowth-related procedures [24]. In this approach, superparamagnetic particles (such as iron oxide-coated polystyrene beads) coated with target organism-specific antibodies are used to isolate the pathogens from the aqueous solution containing a mixed population of microorganisms [25]. Over the past few years, the IMS technique has been reported in combination with polymerase chain reaction (PCR) assays to improve their sensitivity for detection of microorganisms [26,27]. This combination offered a rapid (<7 h) and simultaneous detection of different microorganisms (i.e., *Listeria monocytogenes* [*L. monocytogenes*] and *Listeria ivanovii*) without any enrichment process from the sample [28]. The major limitation with these chemical and physical methods is the interference with surface properties of microorganisms during their recovery and concentration from aqueous solutions.

### 2.3. Immunoassays Approach

Immunoassays are based on the specific interaction between an antigen (i.e., biological contaminant) and an antibody to measure the concentration of microbes in a solution. Different formats and variations of immunological assays have been reported, which may involve the multiple steps of addition and separation of reagents [12,29,30]. In conventional homogeneous immunoassays formats (such as agglutination, immunodiffusion, and turbidemetry), the bound–unbound antibodies are not separated and the formed antibody–antigen complex is easily visible in short incubation times [31]. However, this approach does not facilitate appropriate quantification and detection limits. The heterogeneous immunoassay, in the form of highly specific and sensitive enzyme-linked immunosorbent assay (ELISA), and its varied modifications are now the most prevalent methods on commercial levels for detection of biological contaminants [32]. In this method, unbound antibodies are separated from the bound ones, which helps in improving the detection limit i.e., ranges from 10^3^–10^5^ cfu/mL for whole cells and few ng/mL for proteins/toxins [33]. Although the method is accurate, the direct detection of microbes is not possible; enrichment is required for achieving the required quantification of analytes. The specificity of immunoassays motivates the researchers to explore further modifications in order to achieve higher sensitivity, portability, ease of use, and reusability.

### 2.4. Nucleic Acid-Based Detection

The advancements in biotechnology have resulted in the development of a variety of nucleic acid-based rapid detection platforms for biological contaminants. These methods exhibit higher sensitivity and specificity than culture-based assays, along with drastic reduction in the detection time [34]. The major strategies involved in nucleic acid-based detection of pathogens are hybridization and amplification. Nucleic acid hybridization involves identification of biological contaminants through complementary binding of single-stranded nucleic acid probes (labeled with radioactive or fluorescent compounds) with target organism genes [35,36]. In the presence of complementary target genes in the sample to be tested, the probe forms hydrogen bonds, facilitating its detection through autoradiography or fluorescent signals from the probe. This approach has a limitation in terms of the requirement of a high concentration of target nucleic acid to achieve detectable signals. This limitation can be overcome by nucleic acid amplification-based assays (like PCR), which increase the number of specific target nucleic acid in a sample [37]. PCR can generate multiple copies of a target DNA sequence via different processing steps such as (a) repeated cycles of denaturation of target DNA, (b) annealing of synthetic oligonucleotide primers to denatured strands, and (c) polymerization of target DNA as a template using enzymes. The repeated cycles of PCR result in million-fold amplification of a single copy of target DNA in less than 2 h, thereby eliminating the need for culture enrichment [38]. Cell concentration up to 10^1^–10^6^ cfu/mL has been detected with PCR methods in combination with nucleic acid extraction approaches [39]. Besides the revolutionary nature of PCR in terms high specificity and selectivity, there are still limitations in terms of nonreliable detection and quantification of infectious or viable organisms. Furthermore, its routine use is also not feasible due to requirement of costly reagents, trained personnel, and a very clean working environment.

## 3. Point-Of-Care (POC) Devices for Biological Contaminants

The laboratory-based technologies (such as PCR and ELISA) are not accessible or affordable in resource-limited areas lacking basic infrastructure and/or trained operators. POC devices have emerged to facilitate the on-site detection of different analytes (Table 1) [40,41]. The POC devices are replacing culture-based and other cumbersome commercial methods, which is attributable to widened accessibility to detection, rapid analysis time, and cost effectiveness [42]. Various POCs are majorly developed on microfluidic [43,44,45] or lateral flow systems [46,47,48] using glass; paper; silk fibroin; flexible graphite; or polymer substrates, such as polydimethylsiloxane (PDMS), poly(3,4-ethylene dioxythiophene):poly(styrene sulfonate) (PEDOT:PSS)), poly(methyl methacrylate) (PMMA), and polyethylene terephthalate (PET), etc. Further, the detection approaches are dependent upon fluorescence [49], colorimetry [50], and electrochemistry [51,52,53] to achieve rapid and easy to interpret signals for water safety monitoring.

### 3.1. Paper-Based Assay Methodology

Paper-based devices are attractive candidates for quantitative POC detection of biological contaminants in water as they offer low cost development, simple operation, portability, and robust instrumentation [68]. The main advantages of paper-based sensing devices include (i) high surface area to volume ratio; (ii) efficient adsorption; (iii) biocompatibility; (iv) ease of functionalization for stable immobilization of bioprobes; (v) straightforward sterilization; and (vi) easy disposal via incineration [69,70]. Also, the paper materials are lightweight and easily accessible worldwide with minimal chemical handling required due to convenient storage and transport of reagents within the matrix [71]. A paper-based handheld culture device has been reported for *E. coli* detection in water [54]. The strategy exploited recombinant reported lacZ T4 (carrying β-galactosidase gene) bacteriophage along with bioluminescent (6-O-β-galactopyranosyl-luciferin, Beta-Glo(®)) β-galactosidase indicator substrate to monitor β-galactosidase release post phage-mediated cell lysis, as shown in Figure 2. For detection of contaminants, the filtration of water samples was carried out using membrane filters (having 0.45 µm pore size) to concentrate bacteria followed by incubation of filter papers on nutrient medium-coated paper at 37 °C for 4 h. Next, the bacteriophage and respective indicator substrates were incorporated into the device resulting in cell lysis. The lysed cells released β-galactosidase enzyme, which catalyzed the conversion of chromogenic products into fluorescent product. The fluorescence was detected using a portable luminescence imaging device attached to culture-based paper. The sensor demonstrated a visual detection limit of < 10 CFU/mL within a time period of 5.5 h with high specificity in the presence of other nonspecific bacteria, such as *Enterobacter cloacae, Aeromonas hydrophila,* and *Salmonella Typhimurium.*

The wide characteristics of nanomaterials have been studied for water and wastewater treatment such as catalysis, high reactivity, and adsorption [72,73,74]. Incorporation of advanced nanomaterials with paper-based assays have significantly improved the sensitivity of sensors [75]. A group of researchers has explored graphene oxide (GO), a two-dimensional nanomaterial acting as an excellent quencher to fluorescent quantum dots (QDs) for highly sensitive detection of *Salmonella spp* [55]. The turn ON/OFF testing strips were designed by decorating the test line with (i) CdSe@ZnS QDs as capture probes and (ii) GO as an agent for target revealing. The quenching of fluorescence of the test line occurs in the absence of the target pathogen due to energy transfer between donor Ab-QDs and acceptor GO, resulting in the turn OFF state of the sensor. However, the fluorescence of the test line is restored in the presence of target *Salmonella* spp. due to selective binding of bacteria to Ab-QDs on the test line, which increased the distance (> 20 nm) between donor and acceptor to interrupt the resonance energy transfer. The turn ON state of the test line fluorescence indicated the presence of *Salmonella* with 100 CFU/mL detection limit in milk and water samples. Apart from fluorescence, high surface area and excellent electrical conductivity of graphene have also been investigated for imparting improved sensitivity of paper-based sensors for biological contaminants [76]. The sensing platform employing hierarchical structure of graphene-wrapped copper oxide-cysteine has achieved a detection limit of 3.8 CFU/mL for *E. coli* O157:H7 in fruit juice, milk, and water samples [77].

Ma and coworkers patterned the screen-printed polydimethylsiloxane (PDMS) paper chip-based visible immunoassay for bacterial detection (i.e., *E. coli*) using antibody-immobilized detection zones, where detection antibody-functionalized gold nanoparticles (AuNPs) act as a signal reporter [56]. Following the specific immunological reactions, a silver enhancer solution was added onto the detection zone. AuNPs concentrated on the detection zone catalyzed reduction of silver ions to make the visualized black spots. The silver-stained chips were then scanned using a commercial digital camera and/or scanner to obtain the quantitative results. The calibration of gray value of silver-stained spots with bacterial concentration provided the information on density of *E. coli* in water samples. The sensitivity of the developed strips was found to be comparable to conventional ELISA, but with reduced cost due to lesser consumption of time and reagents, pointing towards its potential for in situ monitoring of water quality.

Another litmus paper-based DipTest has been developed for *E*. *coli* detection in aqueous solutions, in which enzymatic reactions were performed directly on porous cellulose blotting paper [57]. The long strips of paper were coated with a hydrophobic barrier at the top edge, chemoattractant at the bottom edge, and customized chemical reagents immediately below the top edge, forming the reaction zone. During the detection process, the paper strip was dipped in water samples (contaminated with *E*. *coli*), where the bacterial cells were chemotactically attracted towards the paper strip, followed by their ascent towards the reaction zone through capillary wicking. Thereafter, the concentrated *E. coli* at the paper strip (particularly at the reaction zone) reacts with chemical reagents to produce pink to red color, indicating the presence of contamination in the tested samples. The fabricated paper strips have been tested for sensitivity and specificity, on the basis of which their potential for efficient on-site screening of water samples for biological contamination has been established.

Further, efforts have been focused on achieving improvements in sensitivity of the paper-based sensor for pathogen detection via different techniques like electrochemical assay [78,79]. Rengaraj et al. [58] have explored a novel paper-based electrochemical sensor for impedimetric sensing of bacterial contamination (refer to Figure 3). The electrochemical device is composed of screen-printed carbon electrodes fabricated onto hydrophobic paper with incorporated functionality. Concanavalin A has been employed as a recognition element instead of costly antibody probes to reduce the cost of the sensor significantly. Concanavalin exhibits high specificity for mono or oligo-saccharides present on the cell envelope of bacteria. The selection of Concanavalin A as the recognition agent was carried out due to its high selectivity towards oligosaccharides or monosaccharides on bacteria. The electrochemical sensor detected bacteria down to a concentration of 2 × 10^3^ CFU/mL, which is sufficiently low to advocate applicability in real samples. Besides these advancements, there are still several challenges associated with commercial application of paper-based assays such as accuracy, sensitivity, and multiple analytes detection at same time.

Despite the promising properties and applications of paper-based sensors, there are some associated limitations in terms of accuracy, resolution, sensitivity, and multiplexed detection. However, the efforts can be diverted towards patterned fabrication and employment of other advanced detection approaches (for e.g., chemiluminescence and electrochemiluminescence) [68,80].

### 3.2. Microfluidic Detection Platforms for Water Quality Assessment

Microfluidic-based analytical devices are receiving research and commercial orientation for sensing of biological contaminants in environmental samples (i.e., water and food) due to low cost, high throughput, simplicity, minimal sample requirement, rapidity, and on-site applicability [81]. Different works incorporating microfluidic platforms with different analytical approaches have also surfaced for detection of biological contaminants such as PCR [82,83,84], loop-mediated isothermal amplification (LAMP) [85,86], mass spectrometry [87,88], and electrochemistry [89,90]—in particular, a disposable PCR microfluidic device composed of an array of closed reactors preloaded with different pairs of primers for real-time monitoring of multiple water-borne pathogens (coliform bacteria) [91]. PCR detects the genes specific to the organisms through amplification. Capillary flow scheme has been used as single step for PCR mixture loading and isolation of the reactors incorporated to minimize the evaporative losses, reducing the complexity and cost of the device, with a detection limit of 51 CFU/mL. Using the similar microfluidic quantitative PCR (MFQPCR) system, the human viral and enteric bacterial pathogens have been detected down to concentration of 2 copies/μl of cDNA/DNA [92] and 100 cells/liter [93], respectively, in wastewater treatment plants. However, further research efforts are needed to resolve the major limitation of microfluidic PCR (such as bubble generation and reagent evaporation) through appropriate modeling of microfluidic platforms for controlled fluid transfer [94].

Microfluidic paper-based analytical devices (µPADs) have been investigated for the detection of different pathogens in environmental samples, such as *L. monocytogenes* [95,96], *E. coli* [97], *Salmonella* [98], etc. Jokerst and coworkers reported a series of colorimetric spot tests using bioactive paper for sensing of *L. monocytogenes*, *E. coli,* and *S. enteric* in water samples from agricultural lands [59]. The pathogen detection is based on the catalytic conversion of dried compounds predeposited on the µPAD with the enzyme produced by the target bacteria grown in the culture media. For *E. coli* determination, the reaction between chlorophenol red β-galactopyranoside and β-galactosidase results in color transition from yellow to red. Determination of *L. monocytogenes* was facilitated by the production of a blue color on reaction of 5-bromo-4-chloro-3-indolyl-myo-inositol phosphate with phosphatidylinositol-specific phospholipase C. Lastly, the catalytic action of esterase on 5-bromo-6-chloro-3-indolyl caprylate resulted in the formation of a purple colored product, leading to detection of *S. enteric.* The detection limit of 10 CFU/cm^2^ was achieved after 4 to 12 h of enrichment process, which is comparatively faster than culture-based methods. Further, µPAD based on similar colorimetric detection of pathogen *Crronobacter* spp. has been reported [99]. A micro-spot μPAD was fabricated using a polyvinyl chloride (PVC) pad with filter paper. In the presence of *Cronobacter* spp., the reaction of microbe-specific enzymes with chromogenic substrates can be observed in terms of color change, i.e., from colorless to indigo. This color change was measured and co-related to quantify the present analyte with an achievable detection limit of 10 CFU/cm^2^ in 10 h at low-cost.

Park et al. have also demonstrated on-field sensing of *E. coli* in aqueous samples using paper microfluidics and smartphones [60]. A three-channel device was fabricated with different types of anti-*E. coli*-conjugated beads (for both low and high concentrations) and also bovine serum albumin (BSA)-conjugated beads preloaded in two detection and negative control channels, respectively. In the presence of *E. coli* antigens, immune agglutination occurred with specific antibody-conjugated beads preloaded on the paper fibers, while BSA-conjugated beads remained unreacted. The quantification of the extent of immune agglutination was achieved by evaluation of digital images for Mie scatter intensity at an optimized distance and angle using gyroscope installed in smartphones. The assay results were found to be in validation with conventional culture-based methods. The combination of microfluidics and smartphones enabled the on-site, easy, and highly specific detection of *E. coli* with sufficiently high sensitivity with much lesser analysis time of 90 s. Microfluidic chip-based electrochemical sensors have also been intensively explored for biological contaminant detection in food and water samples with high sensitivity and rapid response [100,101]. The progress in the area of fabrication and microelectronics has facilitated the incorporation of electrochemical assay into microfluidic platforms with minimal sample, time, and cost requirements [102]. In microfluidic chip-based electrochemical sensors, the probe composed of either bare conductive substrate or biorecognition molecules immobilized the substrate on the electrode surface, as well as the analyte to be detected; the analyte brings the measurable changes in electrochemical properties of the electrode [103]. Altintas et al. [61] fabricated a fully automated and real-time electrochemical detection platform for pathogens, refer to Figure 4. The custom-designed biosensing platform offered sensitive quantification of *E. coli* with a detection limit of 50 CFU/mL. The regeneration ability of sensor surface and high specificity towards E. coli detection in the presence of other pathogens (i.e., *Shigella*, *S. spp*., *S. aureus*, and *S. typhimurium*) make it a promising tool for pathogen monitoring in water samples.

Kim and coworkers have developed a label-free impedance-based positive dielectrophoretic microfluidic detection platform for assessment of *E. coli* contamination in drinking water [62]. The chip is composed of a cell-focusing electrode for sorting of bacterial cells from water and a sensing electrode for subsequent capture and sensing of bacterial analyte. The binding of bacterial cells resulted in the change in resistance to charge transfer at the electrode surface and these impedance changes were then used to enumerate *E. coli* bacterial cells down to the detection limit of 300 CFU/mL in less than 1 min. The major limitation with label-free detection platforms is lack of specificity, which can generate false positive results on the attachment of nonspecific microorganisms to the electrode surface. Therefore, a group of researchers have used specifically tagged target biological contaminants prior to impedance-based electrochemical detection [63]. The microfluidic chip-based immunosensor is composed of anti-*E. coli* O157:H7 antibodies immobilized on to nanoporous alumina membrane. The explored strategy resulted in improved sensitivity with a relatively lower detection limit of 100 CFU/mL and negligible interference from nontarget bacteria. The integration of microfluidic chips with different analytical techniques is contributing towards the progress of microfluidics for pathogenic microbe detection in human body fluids, food, water, and air.

### 3.3. Lateral Flow Devices for Contaminant Detection

Lateral flow devices are one of the most commercialized POC testing methods, attributable to low-cost, easy interpretation, high specificity, and portability for detection and quantification of different analytes in complex mixtures. In the modern era, lateral flow assay-based rapid tests have been developed and widely accepted by regulatory authorities for screening of diseases [104], chemicals [105], pathogens [106], water pollutants [107], and toxins [108]. Lateral flow assays offer one-step detection, without the involvement of washing steps, in a simple manner, with relatively short development time, low-cost production, and low sample volume requirements. A novel lateral flow device for ultrasensitive and highly specific multiplexed sensing of pathogenic as well as nonpathogenic *E. coli* based on intracellular enzymatic activity of β-glucuronidase (GUS) or β-galactosidase (β-GAL) has been reported on bioactive strips [64]. The test strips were printed with sol-gel-derived silica ink using ink-jet printing technology, entrapping chlorophenol red β-galactopyranoside (CPRG), 5-bromo-4-chloro-3-indolyl-β-d-glucuronide sodium salt (XG), or both and FeCl_3_ at different zones. For detection, the sample was lysed and transported through the metal salt zone to the different substrate zones via lateral flow. On reaching the substrate, enzymatic hydrolysis of the substrate is initiated by the intracellular enzymes, resulting in production of blue colored and yellow to red-magenta product for XG hydrolysis by GUS, indicative of nonpathogenic *E. coli,* and CPRG hydrolysis by β-GAL, indicative of total coliform bacteria, respectively. The immunomagnetic nanoparticles were also employed for further selective preconcentration, which helped in achieving a very low detection limit of 5 CFU/mL and 20 CFU/mL for *E. coli* O157:H7 and *E. coli* BL21, respectively within 30 min with high specificity. In this work, the inclusion of the bacterial culturing step allowed detection of 8 CFU/ 100 mL *E. coli* cells in 8 h.

The major challenge of POC devices in food and water safety applications is lack of multiplexed detection, with reduced risk of cross-reactivity among various target analytes. Chen et al. [65] reported the fabrication of lateral flow test strips containing multiple sensing zones for detection of *Pseudomonas aeruginosa (P. aeruginosa)* toxin genes (i.e., ecfX, ExoU, and ExoS) in drinking water samples. The different capturing molecules were embedded into multiple detection zones to detect anti-hex (homeobox protein), anti-FITC (fluorescein isothiocyanate), and anti-digoxin antibody-tagged LAMP amplicons. Another study on the multiplexed detection of different pathogens using a 10-channel lateral flow device has been reported through upconverting phosphor technology-based fluorescence [109]. For this, 10 individual paper strips were each immobilized with antibodies specific to different pathogens, including *S. typhimurium*, *E. coli* O157:H7, *Vibrio cholera O1*, *V. cholera O139*, *V. parahaemolyticus, S. paratyphi A*, *S. paratyphi B*, *S. paratyphi C*, *S. choleraesuis*, and *S. enteritidis.* Further, the integration of individual strips was carried out by a single lateral flow device. The multiplexed sensor achieved detection limits in the range of 10^4^–10^5^ CFU/mL for different pathogens without the need of sample enrichment. The proposed device holds greater promise for rapid safety assessment of food and water resources.

The integration of lateral flow assay with smartphones is paving the way for simultaneous detection of multiple analytes, depending upon the different analytes (such as ions, bacteria, and small molecules) and recognition element reactions in the same sample, refer to Figure 5 [66]. The detection probe consisting of nanoparticles with different emission bands can resolve the issue of crossover reactions from multiple analytes. However, some fundamental features of lateral flow assays must be improved in terms of reproducibility and accuracy comparable to laboratory-based detection systems.

Researchers are exploring various ways to integrate different analytical techniques into one system to develop paper-based sample-to-answer devices for qualitative and quantitative determination of biological contaminants. For instance, LAMP, nucleic acid testing (NAT), and colorimetric detection methods have been orchestrated into a robust, simple, four-layered lateral flow device [67]. Flinders Technology Associates (FTA) cards and glass fibers have been used as extraction and amplification substrates for nucleic acid, respectively. The separation of paper layers by polyvinyl chloride (PVC) acted as valves for controlling the sample transportation from extraction to amplification and further to lateral flow zones. The fully integrated device was able to detect *E. coli* down to concentration as low as 10−100 CFU/mL, supporting its efficient use in safety monitoring of food and water samples. Overall, lateral flow assays provide cheap, rapid, and easy point-of-care testing of different analytes, but, require improvements in a variety of areas such as signal amplification, precise sample volume accuracy, limit of sensitivity, operation without good antibody preparation, dependence of analysis time on nature of sample, pore obstruction due to matrix obstruction, and simple procedure for sample pretreatment [110].

## 4. Conclusion and Future Prospects

In summary, this review presented an overview of recent advancements in the area of developing devices (from conventional approaches to POC) to detect biological contaminants such as pathogens, their components, and toxic secretions, largely in water samples. The presence of these contaminants is responsible for a variety of acute and chronic health hazards, along with major economic losses. Therefore, the qualitative and quantitative estimation of biological contamination is of esteemed importance. Conventionally, culture-based microbiological assays, molecular detection platforms, ELISA, separation, and filtration techniques are used on commercial levels. These methods are specific and accurate, but suffer from limitations of long analysis time, high cost, requirement of heavy instrumentation and trained personnel, and nonapplicability for on-site detection. Therefore, portable POC devices are being explored to enable highly sensitive, specific, and personalized detection of environmental contaminants. The majority of POC devices employ either antibody–antigen interaction-based immunoassays, or nucleic acid-based hybridization and amplification assays. The most common transduction techniques incorporated in these devices are visible colorimetry, fluorescence, or electrochemical detection. Further, it has been elucidated that the POC devices are fabricated on paper, microfluidic chip, or lateral flow platforms. 

While POC devices are being extensively studied, several formidable challenges require addressing for their translation into commercial field applications. Efforts are required to achieve enhanced sensitivity, multiplexing, portability, and ease of quantification in POC devices for water and food safety assessment. For improved sensitivity, signal amplification using advanced nanostructures exhibiting intrinsic extraordinary optoelectronic and catalytic properties can be integrated into chip- and paper-based devices. Incorporation of smartphones and other similar readers can facilitate portability, easy understanding of results, and wide real field applicability. Also, the exploration of methods of fabricating POC devices with multiplexed detection requires considerable scientific attention in order to reduce cost and assay time, and increase assay productivity. The integration of multiple key detection methods into a single system is an emerging platform for simultaneous detection of multiple analytes. However, the area is still at its infancy and requires research orientation. The advent of such improvement in POC devices will aid in the achievement of water and food safety analysis and quality control in an easy, rapid, inexpensive, and effective manner.

## Figures and Tables

**Figure 1 sensors-19-04476-f001:**
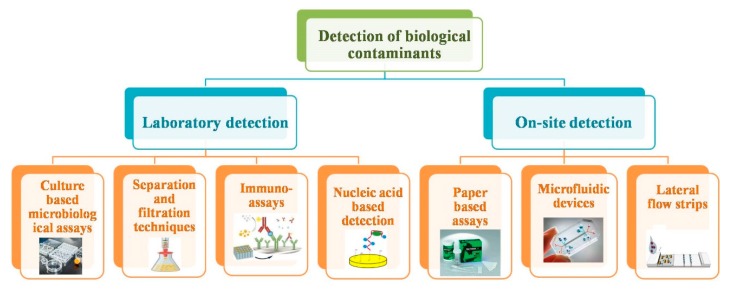
Detection methods for biological contaminants.

**Figure 2 sensors-19-04476-f002:**
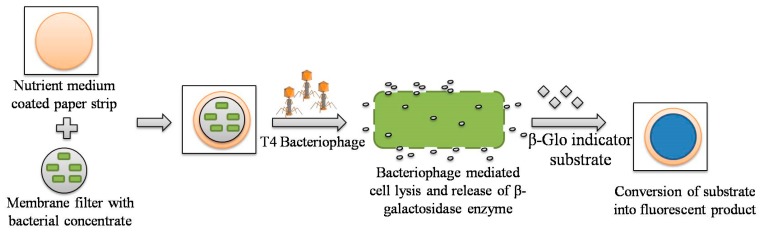
Principles for the detection of *E. coli* bacterial cells using a paper-based handheld culture device.

**Figure 3 sensors-19-04476-f003:**
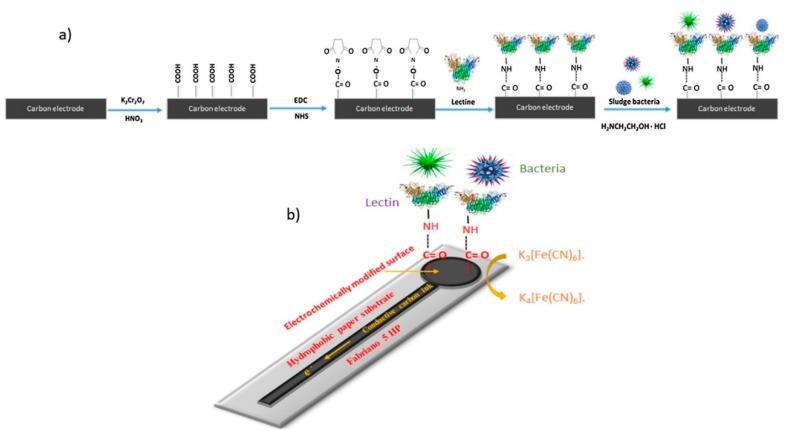
Schematic representation of paper-based biosensors for bacterial contaminant sensing from water. (**a**) Multiple steps of modification of the electrode surface, followed by a working principle of bacteria sensing; (**b**) layout of final paper-based screen-printed carbon electrode (reprinted with permission from [58]).

**Figure 4 sensors-19-04476-f004:**
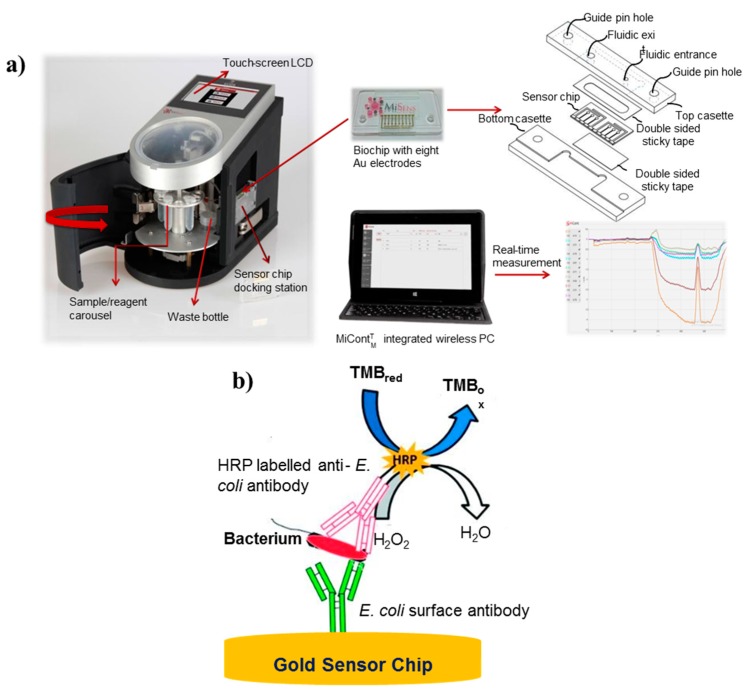
Schematic illustration of microfluidic-based electrochemical biosensor. (**a**) Fully automated detection system and (**b**) assay for pathogen detection (reprinted with permission from [61]).

**Figure 5 sensors-19-04476-f005:**
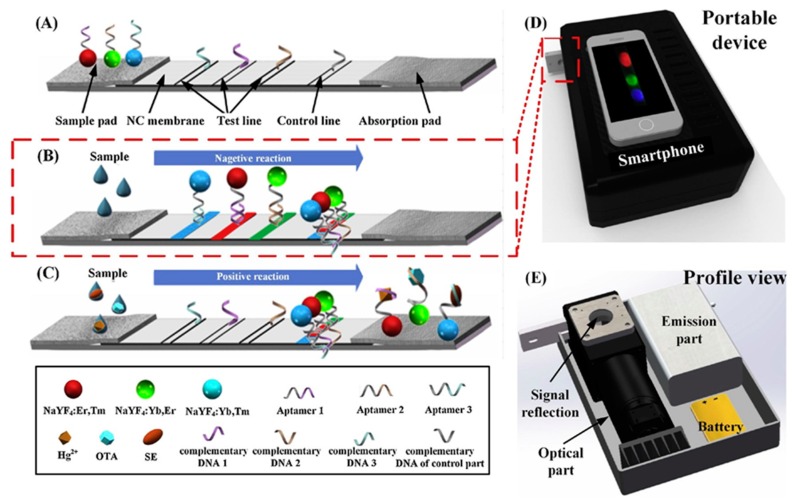
Schematic representation of lateral flow aptamer assay for simultaneous sensing of multiple analytes. (**A**) Structure of lateral flow aptamer assay, (**B**) hybridization of upconversion nanoparticles probe with complementary DNA in the absence of target analytes, (**C**) decrease in fluorescence of test zones due to aptamer bonding with target analytes, (**D**) display of detection results using smartphone-based portable device, and (**E**) schematic view of smartphone-based portable device (reprinted with permission from [66]).

**Table 1 sensors-19-04476-t001:** List of various platforms available for on-site detection of biological contaminants.

S. No.	Type	Analyte	Substrate	Transduction Platform	Detection Limit (CFU/mL)	Analysis Time	Cost	Lifetime	Ref.
1	Paper-based	*E. coli*	Filter paper	Fluorescence	10	5.5 h	Moderate	Single use	[54]
2		*Salmonella* spp.	CdSe@ZnS QDs decorated paper strips	Fluorescence	3.8	-	Moderate	Single use	[55]
3		*E. coli*	AuNP decorated PDMS paper chips	Optical immunoassay	57	-	High	Reusable	[56]
4		*E. coli*	Litmus paper	Colorimetry DipTest	2 × 10^5^ to 4 × 10^4^	-	Very low	Single use	[57]
5		Bacterial Contaminant	Screen printed carbon electrode	Electrochemical impedance	2 × 10^3^		Moderate	Reusable	[58]
6	Microfluidic	*L. monocytogenes, E. coli, S. enteric*	Color-producing compounds deposited on µPAD	Colorimetry	10	4−12 h	Moderate	Reusable	[59]
7		*E. coli*	Paper fibers	Gyroscope installed in smartphone	10	90 s	Moderate	Reusable	[60]
8		*E. coli*	AuNP-coated biochips	Cyclic voltammetry and amperometry	50	8 min	High	Reusable	[61]
9		*E. coli*	Dieletrophoretic microfluidic chip	Electrochemical impedance	300	< 1 min	High	Reusable	[62]
10		*E. coli*	Nanoporous alumina membrane	Electrochemical impedance	100	-	Moderate	Reusable	[63]
11	Lateral flow	*E. coli*	Sol-gel-derived silica ink-coated test strips	Colorimetry	5	30 min	Moderate	Reusable	[64]
12		*Psuedomonas aeruginosa*	AuNP-conjugated nitrocellulose membrane	Visual detection	20	50 min	Low	Single use	[65]
13		*Salmonella*	Upconverting nanoparticles-coated paper strips	Colorimetry	85	30 min	High	Single use	[66]
14		*E. coli*	Flinders Technology Associates (FTA) cards and glass fibers	Colorimetry	10−100	-	High	Reusable	[67]

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
