# Peer review of "Point-of-Care Strategies for Detection of Waterborne Pathogens"

_sensors, 2019, doi:10.3390/s19204476_

Round 1
Reviewer 1 Report
Authors review the recent developments in point of care devices for microorganism detections in water.
However, about 30% of the references were used to review many non-relevant contents (i.e. other detection methods); therefore, lots of state-of-the-art developments are not included; for example, the microfluidic PCR.
This feature articles "Anal. Chem. 2018, 90, 5512−5520" (actually in the reference of this manuscript) will provide some guidance for improvements
Also, there are many missing pieces in this review such as the pros and cons of different POC technologies and the challenges (the real challenges with judgments not just randomly listing something like sensitivity, multiplexing, portability, and ease of quantification, which already been addressed by many lab-based prototypes).
Based on these reasons, I suggest that this review should not be published in high prestige journal like Sensors before significant improvements.
Author Response
Response to reviewer
Manuscript ID: sensors-610614
Type of manuscript: Review
Title: Point-of-care strategies for biological contaminants detection in water
Author remark. Authors do appreciate valuable comments of reviewers. Authors have carefully addressed every query in revised manuscript.
Reviewer #1
Response to Reviewer
Comment 1. Authors review the recent developments in point of care devices for microorganism detections in water. However, about 30% of the references were used to review many non-relevant contents (i.e. other detection methods); therefore, lots of state-of-the-art developments are not included; for example, the microfluidic PCR.
Response. As per the reviewer’ comment, the recent advancements in POC technologies have been explored in terms of paper-based assays, microfluidic platforms, and lateral flow devices. Appropriate changes have been done throughout the manuscript to clear the state-of-the-art of recent advancements, especially the microfluidic PCR (Page No. 11, line 272 onwards).
Comment 2. This feature articles "Anal. Chem. 2018, 90, 5512−5520" (actually in the reference of this manuscript) will provide some guidance for improvements
Response. As per the reviewer’ suggestion, the suggested article is reviewed to get guidance for improvements. Appropriate changes have been done throughout the manuscript (highlighted with yellow color).
Comment 3. Also, there are many missing pieces in this review such as the pros and cons of different POC technologies and the challenges (the real challenges with judgments not just randomly listing something like sensitivity, multiplexing, portability, and ease of quantification, which already been addressed by many lab-based prototypes).
Response. As per the reviewer’ suggestion, the pros and cons of different POC technologies are discussed at appropriate places in the manuscript along with emphasis on challenges faced by particular POC technology.

Reviewer 2 Report
The review presents an overview that recapitulates reported water contaminants detection platforms (from conventional approaches to POC) to detect biological contaminants in water samples.
The work is well written, just some grammar and contextual spelling have to be addressed in Line 36-37, Line 51-52 and Line 157.
Table 1 The first two references have different referencing styles. Also, the size of the content in the table has to be adjusted to avoid cut words
Author Response
Response to reviewer
Manuscript ID: sensors-610614
Type of manuscript: Review
Title: Point-of-care strategies for biological contaminants detection in water
Author remark. Authors do appreciate valuable comments of reviewers. Authors have carefully addressed every query in revised manuscript.
Reviewer #2
Response to Reviewer
Comment 1. The review presents an overview that recapitulates reported water contaminants detection platforms (from conventional approaches to POC) to detect biological contaminants in water samples.
The work is well written, just some grammar and contextual spelling have to be addressed in Line 36-37, Line 51-52 and Line 157.
Response. We are highly thankful to the reviewer for his valuable suggestion. We have revised the specified lines as well as whole manuscript keeping in view the reviewer’ suggestion (highlighted with yellow color).
Comment. The first two references have different referencing styles. Also, the size of the content in the table has to be adjusted to avoid cut words.
Response. As per the reviewer’ suggestion, Table 1 is revised (refer to Page 6, line 178).

Reviewer 3 Report
This manuscript reviews the point-of-care (POC) strategies for detection of waterborne pathogens. The manuscript is interesting and well-written. However, the following comments need to be addressed before the manuscript can be accepted for publication.
The manuscript only focuses on detection of waterborne pathogens using POC devices, so I would suggest the authors to change the title to “Point-of-care strategies for detection of waterborne pathogens”. In order to have a fuller comparison between the POC devices, it would be very helpful to include the required assay time for each device in Table 1. Line 213: The authors should add “detection” after “where”. Lines 214 – 215: “Following the specific immunological reactions, AuNPs concentrated on the paper chip catalyzed the silver ions reduction.” This sentence should be revised as follows: "Following the specific immunological reactions, a silver enhancer solution was added onto the detection zone. AuNPs concentrated on the detection zone catalyzed reduction of silver ions to make the visualized black spots." The following relevant works should be cited and discussed: A novel, simple and low-cost paper-based analytical device for colorimetric detection of Cronobacter spp. 2018. Analytica Chimica Acta, 1036: 80-88. Emerging point-of-care technologies for food safety analysis. 2019. Sensors, 19(4): 817.Author Response
Response to reviewer
Manuscript ID: sensors-610614
Type of manuscript: Review
Title: Point-of-care strategies for biological contaminants detection in water
Author remark. Authors do appreciate valuable comments of reviewers. Authors have carefully addressed every query in revised manuscript.
Reviewer #3
Comments and Suggestions for Authors
This manuscript reviews the point-of-care (POC) strategies for detection of waterborne pathogens. The manuscript is interesting and well-written. However, the following comments need to be addressed before the manuscript can be accepted for publication.
Author Remark.
Comment 1. The manuscript only focuses on detection of waterborne pathogens using POC devices, so I would suggest the authors to change the title to “Point-of-care strategies for detection of waterborne pathogens”.
Response. We are highly thankful to the reviewer for his valuable suggestion. The title has been changed as per the suggestion.
Comment 2. In order to have a fuller comparison between the POC devices, it would be very helpful to include the required assay time for each device in Table 1. Line 213: The authors should add “detection” after “where”. Lines 214 – 215: “Following the specific immunological reactions, AuNPs concentrated on the paper chip catalyzed the silver ions reduction.” This sentence should be revised as follows: "Following the specific immunological reactions, a silver enhancer solution was added onto the detection zone. AuNPs concentrated on the detection zone catalyzed reduction of silver ions to make the visualized black spots."
Response. As per the reviewer’ suggestion, all the mentioned changes have been incorporated and highlighted in the manuscript.
Comment 3. The following relevant works should be cited and discussed: A novel, simple and low-cost paper-based analytical device for colorimetric detection of Cronobacter spp. 2018. Analytica Chimica Acta, 1036: 80-88. Emerging point-of-care technologies for food safety analysis. 2019. Sensors, 19(4): 817.8
Response. As per the reviewer’ suggestion, the above-mentioned relevant works have been cited and discussed in the manuscript at the appropriate place.

Round 2
Reviewer 1 Report
The authors have addressed most of my comments.
I have one more comment that authors should comprehend table one by comparing more technologies introduced in their reference (Lateral-flow devices and many microfluidic devices are not in the table) as well as adding more factors in comparisons e.g. cost (high or low), and lifetime (one-time use? multiple tests ?).
This paper is recommended for publication after this comment is addressed.
Author Response
Response to Reviewer Comments
I have one more comment that authors should comprehend table one by comparing more technologies introduced in their reference (Lateral-flow devices and many microfluidic devices are not in the table) as well as adding more factors in comparisons e.g. cost (high or low), and lifetime (one-time use? multiple tests ?).
Ans.] As per the learned reviewer’s suggestion, Table 1 is comprehended by comparing more technologies, i.e. Lateral-flow devices, microfluidic devices and other factors are also included for comparison (Page No. 6, line 180 onwards).